# General, Vaccination, Navigational and Digital Health Literacy of Students Enrolled in Different Medical and Health Educational Programs

**DOI:** 10.3390/healthcare12090907

**Published:** 2024-04-26

**Authors:** Gaukhar Kayupova, Aliya Takuadina, Zhanerke Bolatova, Zhaniya Dauletkaliyeva, Nurbek Yerdessov, Karina Nukeshtayeva, Olzhas Zhamantayev

**Affiliations:** 1School of Public Health, Karaganda Medical University, 40 Gogol Street, Karaganda 100008, Kazakhstan; kayupovag@qmu.kz (G.K.); bolatovazhanerke93@gmail.com (Z.B.); dauletkalievaz@qmu.kz (Z.D.); erdesov@qmu.kz (N.Y.); nukeshtaeva@qmu.kz (K.N.);; 2Department of Informatics and Biostatistics, Karaganda Medical University, 40 Gogol Street, Karaganda 100008, Kazakhstan

**Keywords:** health literacy, medical, student, digital, navigational, vaccination, Kazakhstan

## Abstract

Evaluating prospective graduates’ health literacy profiles before they enter the job market is crucial. Our research aimed to explore the health literacy levels of medical and health students by assessing their ability to obtain health-related information, understand healthcare systems, use e-health, and be informed about vaccination as well as to explore the factors associated with health literacy. Short versions of the HLS19-Q12 were used for a cross-sectional survey that was carried out among 1042 students enrolled in various medical and health educational programs at three medical universities in Kazakhstan between September and November of 2023. Additionally, instruments such as Digital Health Literacy (HLS19-DIGI), Navigational Health Literacy (HLS19-NAV), and Vaccination Health Literacy (HLS19-VAC) were employed. The score of General Health Literacy was 88.26 ± 17.5. One in eight students encountered difficulties in Vaccination Health Literacy. Despite overall high health literacy, Navigational Health Literacy posed challenges for all students. The Public Health students exhibited the highest General Health Literacy (91.53 ± 13.22), followed by students in Nursing, General Medicine, other educational programs (Dentistry and Biomedicine) and Pharmacy. Financial constraints for medication and medical examinations significantly influenced health literacy across all types of individuals. Since comprehensive health literacy instruction or interventions are still uncommon in the curricula, it seems reasonable to develop and incorporate appropriate courses for medical and health educational programs.

## 1. Introduction

Health literacy (HL) is the level of capacity an individual has to find, comprehend and apply information and resources to guide health-related choices and activities for oneself and others [1,2]. The WHO states that HL entails reaching a degree of understanding, self-assurance and personal abilities to take action to enhance one’s own and the public’s health by modifying one’s own lifestyle and living circumstances [3]. Consequently, because HL makes health information more accessible and easier to understand, it is crucial for empowerment. Employing HL best practices can increase health equality and foster trust. An essential component of someone’s desire to practice self-care and health-promoting habits is trust. Limited HL is recognised as a serious problem with both health and financial implications [4,5]. Additionally, HL plays a key role in efficient patient–provider communication. The WHO has identified the problem of limited HL in the population as an important public health concern [6].

The environment for increasing population literacy and empowering people to make decisions about their own health should constitute the cornerstone of health policy. Thus, as a part of the Republic of Kazakhstan’s 2020–2025 state program for healthcare development, many initiatives are being implemented to raise public HL and influence public health-related behaviour [7]. 

Assuming that HL primarily consists of a functional assessment of reading and numeracy skills, the majority of health providers, if not all of them, will have high HL. Since health workers serve as the population’s reference points within the healthcare system, this helps patients’ HL as well. However, this assumption might not be accurate given the increasing acknowledgement of HL’s multidimensional character [8]. 

HL and communication quality and continuum are critical for the delivery of patient-centered care and have been recognised by researchers as important points for the training of healthcare professionals [9]. However, a limited number of studies have aimed to assess HL comprehensively in both healthcare professionals and medical students [10,11,12,13,14], and these studies have reported conflicting results. Thus, a number of studies report limited levels of HL and knowledge gaps in this area among both healthcare providers and medical students, requiring close attention [15,16,17,18,19,20,21]. However, it has been reported that the level of HL among medical students remains high and significantly exceeds the average level of students in other fields of study [22,23]. Given that healthcare professionals usually spend more than four years in college learning about their field, it seems reasonable to evaluate prospective graduates’ HL profiles prior to their entry into the job. It is the duty of universities to educate future physicians so that they can recognise and address the healthcare needs of the people they will eventually serve [24]. Understanding the HL level of medical students and addressing any gaps provides a mechanism for preparing professionals who can effectively provide healthcare [25]. 

Despite the fact that there are a number of studies assessing HL in the general population in Kazakhstan [26,27,28,29,30], research in the field of assessing HL among medical students in Kazakhstan is limited and is represented by one study in the western region of Kazakhstan. This study of HL among medical students reported a significant prevalence of adequate levels of HL among students [31]. 

The HLS19-Q12 general HL assessment tool, along with additional Navigational, Vaccination and Digital Health Literacy assessment packages, were developed as part of the “HLS19—the International Health Literacy Population Survey 2019–2021” project of M-POHL [32,33,34]. The instrument showed an excellent level of consistency across all participating countries, and the unidimensionality of the scale was confirmed by CFA (confirmatory factor analysis) and Rasch models, with acceptable results for all the scales [35]. In Kazakhstan, this assessment tool was previously used as part of a study by a consortium on the cultural adaptation of the questionnaire among the Russian-speaking population of five countries [28]. 

In our study, we aimed to explore the health literacy levels of medical and health students in Kazakhstan, examining their capacity to access health-related information, navigate healthcare, engage with e-health, and be aware of vaccination and the potential influencing factors.

## 2. Materials and Methods

### 2.1. Participation and Data Collection

A cross-sectional survey was conducted between September and November of 2023 in a representative sample of 1042 students of various educational programs from three medical universities in Karaganda, Astana and Almaty cities, Kazakhstan. We used the stratified sampling method to select participants, ensuring representation from various educational programs such as General Medicine, Public Health, Nursing, Pharmacy, Dentistry and Biomedicine. Stratified sampling involves dividing the total population into distinct subgroups or ‘strata’ (in our case, based on educational programs) and then randomly selecting participants from each subgroup. This method helps to reduce sampling error and ensures that specific groups are not underrepresented. Within each stratum (educational program), participants were selected using simple random sampling. This was done through randomised number generation corresponding to student enrollment lists. Two educational programs, Dentistry and Biomedicine, were combined into a new category called “Others”. The eligibility criterion included being registered as a student at one of three universities (Karaganda Medical University, Astana Medical University or Asfendiyarov Kazakh National Medical University). The survey was available in print and online. The electronic version of the questionnaire was available online on the Google platform, with access offered via QR code. The study involved students from the first to the fifth years of the universities. 

The study used short adapted and validated forms of the HLS19-Q12, which were developed based on the HLS-EU instrument in the frame of the project “HLS19 the International Health Literacy Population Survey 2019–2021 of M-POHL”. It is a 12-item short-form questionnaire that belongs to the HLS19 family of HL measurement tools [32,33,34,35]. The tool was designed to measure the HL of the general population and encompasses a broad public health perspective. Additionally, optional instruments, such as Digital Health Literacy (HLS19-DIGI), Navigational Health Literacy (HLS19-NAV), and Vaccination Health Literacy (HLS19-VAC), were used. A total of 31 correlates were employed for all the surveys, with the HLS19-DIGI and HLS19-VAC utilizing additional correlates. 

The HLS19-DIGI includes skills for effectively searching for, accessing, comprehending, evaluating, verifying and utilising online health information. It also involves the capacity to construct and articulate inquiries, viewpoints, ideas or emotions while utilising digital technologies. A scale consisting of eight questions was created to assess patients’ ability to manage health information in digital formats, together with two items specifically targeting the interactive utilisation of digital devices [36]. The HLS19-NAV consists of twelve items that assess individuals’ self-perceived challenges in accessing, comprehending, evaluating and utilising navigation-related information, mainly for specific activities at the social and organisational levels of navigating healthcare services [37]. 

The HLS19-VAC indicates the ability of individuals to acquire, comprehend and assess information connected to immunisation to make well-informed decisions about becoming vaccinated. Furthermore, it included a survey regarding personal vaccination behaviour over the past five years, four questions concerning personal confidence in vaccinations, three questions regarding misconceptions about potential vaccination risks, and one question regarding the risk of contracting a preventable disease if not vaccinated [38]. 

We obtained permission to use the tools for study from the International Coordination Center (M-POHL network).

### 2.2. Ethical Considerations

The Local Bioethics Commission of the Medical University of Karaganda, dated 11 October 2022 (Protocol 1), approved the study. Participants were aware that participation was voluntary and that they were free to opt in or out of the study at any point in time. The study was anonymous, and instruction on how to complete the questionnaire was provided. Informed consent was obtained from the participants before the study was conducted.

### 2.3. Measures and Statistical Analysis

Generally, the first part of the questionnaire was about correlates. The correlates included questions such as gender, age, region, settlement type, faculty, study course, education level, height, weight, parental education level, any issues with health, health topic, presence of emergency medical skills, financial status, close people whom they could trust, habits such as smoking, drinking alcoholic drinks, physical activity, healthy nutrition points, estimation of health, presence of chronic diseases, number of emergency medical care facilities, visiting doctors, general practitioners, and number of people staying in hospitals and day care centers during the year. 

For the questionnaires, answer options were provided on a 4-point Likert-type scale ranging from 1 (very difficult) to 4 (very easy). The score was determined by summing the response categories “very easy” or “easy” across the items and then standardising the raw score to a range of 0 to 100. Greater scores in the sums imply greater proficiency in participants with high-level HL. 

HL levels were calculated using the following formula: (Number of “easy” or “very easy” responses/Number of valid responses) × 100. The criteria for classifying HL were determined as follows: less than 50 points (inadequate); between 50 and 66.66 points (problematic); between 66.67 and 83.33 points (adequate); and above 83.34 points (excellent). In our study, we consolidated HL levels into two main levels. This decision was due to the low frequency of problematic HL levels. As a result, the levels of “inadequate”, “problematic”, “adequate” and “excellent” were combined. This approach allowed us to optimise the research results [39].

The statistical analysis was performed with R-studio software version 1.2.5033 (Posit, PBC, Vienna, Austria). A two-sided *p*-value < 0.01 was considered to indicate statistical significance. 

A descriptive analysis was conducted. For assessing statistical differences between the General Health Literacy scores between the different educational programmes, the ANOVA test and Tukey’s Honestly Significant Difference (Tukey’s HSD) post hoc test for pairwise comparisons was perfomed. Both-direction stepwise linear regression was perfomed to determine factors influencing General, Digital, Vaccination and Navigational Health literacy. The dependent variables in the models were General, Vaccination, Digital and Navigational Health literacy. The independent variables are presented and described in Table 1, Table 2, Table 3 and Table 4. By adding important variables and eliminating unimportant ones, the method in both-direction stepwise regression optimises the model by combining forward and backward stages. The optimised regression models in this study are as follows:

General Health Literacy = Age + Living conditions + Financial ability to afford medical examination + Health assessment + Digital Health Literacy Score + Navigational Health Literacy Score + Vaccination Health Literacy Score + Field of Medical Education.

Digital Health Literacy = Area of origin + Financial ability to afford medical examination + General Health Literacy Score + Navigational Health Literacy Score + Vaccination Health Literacy Score.

Navigational Health Literacy = Age + Financial ability to afford medical examination + Social status + Health assessment + Physical activity + General Health Literacy Score + Digital Health Literacy Score + Vaccination Health Literacy Score.

Vaccination Health Literacy = Gender + Age + Living condition + Smoking + General Health Literacy Score + Digital Health Literacy Score + Navigational Health Literacy + Field of Medical Education.

## 3. Results

### 3.1. Demographic and Socioeconomic Characteristics of the Sample

The gender distribution was as follows: 26.2% male and 73.8% female (Table 1). The sample had a greater representation of females. The participants’ mean age was 22.2 ± 7.9 years. The majority of the participants were from urban areas (62.5%), and 37.5% were from rural areas. A significant portion of the students resided in dormitories (35.0%), while the other part had living conditions fairly balanced between renting an apartment and living with parents. 

In terms of parents’ education level, 31.7% of fathers and 37.9% of mothers had at least a bachelor’s degree. A greater percentage of participants’ fathers (32.7%) had completed only secondary education than did participants’ mothers (26.0%). The combined average for master’s and Ph.D. levels attained for fathers was approximately 4.8%, and for mothers, it was approximately 7.25%. 

The dataset predominantly comprises participants living in full families, with 88.1% responding affirmatively. A considerable 83.9% of participants found it easy to afford medicines, while 16.0% faced challenges in this regard. Although a majority (72.6%) could easily afford health examinations, a notable portion (27.4%) experienced financial constraints in this aspect. On average, participants reported a relatively high social and financial status, with means of 7.8 ± 1.8 and 7.1 ± 1.8 out of 10, respectively. 

### 3.2. Health- and Health Behaviour-Related Sample Information

By examining the health- and health behaviour-related characteristics of students across various education fields (Table 2), we observed distinct patterns in BMI (body mass index) distribution, prevalence of health problems among family members, self-reported health assessments, activity limitations, and prevalence of chronic diseases. In the total sample, the majority of the students were in the normal weight category (60.7%). Approximately one out of the four Nursing students were overweight. The prevalence of health problems among family members varied among students from different educational fields. Public Health students reported the highest percentage of such issues (39.2%), followed by General Medicine students (30.4%). The General Medicine students reported the highest percentage of “good” health (79.0%), while the Public Health students had a relatively lower percentage (65.3%) of self-reported health assessments. Moreover, only 31.9% of all the students reported no activity limitations due to health problems. Additionally, Nursing and Public Health students reported higher percentages of chronic diseases (29.5% and 28.8%, respectively) than did General Medicine (21.8%) and Pharmacy students (25.4%). 

The health-related behaviours of the participants reflected the diverse natures of smoking, alcohol consumption, physical activity, and healthy food habits within the sample (Table 3). Most participants reported never smoking (93.7%), with a small percentage indicating occasional (2.4%) or regular (1.9%) smoking. Former smokers accounted for 0.4%. A significant proportion of participants reported never consuming alcohol (92.4%), while 7.1% reported occasional consumption. Regular alcohol consumption and N/A responses were minimal, each accounting for 0.2%. Participants engaged in varying levels of physical activity, with the majority reporting less than once a week (21.6%) or at least once (11.9%) per week. Notably, almost one out of five students reported not engaging in any physical activities at all (19.9%). The other categories ranged from 4.1% to 12.3%. A diverse pattern of healthy food consumption was observed, with the highest percentages occurring for occasional (20.7%) and once-a-week (14.8%) consumption. 

### 3.3. Health Literacy and Domain Assessment

The Mean General Health Literacy score was 88.26 ± 17.5 (Table 4), with a median score of 91.7. The majority of participants exhibited adequate General Health Literacy (95.0%), while a small proportion faced problematic HL challenges (5.0%). The average Digital Health Literacy was 85.79 ± 22.8, with most participants exhibiting adequate Digital Health Literacy (88.4%), while a notable proportion experienced problems (11.6%). The average Navigational Health Literacy score was 84.0 ± 27.4. Like in the other domains, a great share of the participants had adequate level (84.5%), while a considerable proportion of the participants reported Navigational Health Literacy issues (15.5%). Regarding the Vaccination Health Literacy test, our participants had an average score of 87.9 ± 25, with a median score of 100. One out of eight students faced challenges in this domain. 

For General Health Literacy, the percentage of adequate literacy was highest for Public Health (96.8%), followed by Nursing (96.59%), Other fields (95.49%), General Medicine (94.45%) and Pharmacy (93.28%) students. The percentage of problematic literacy was correspondingly low in these fields. 

In the case of Digital Health Literacy, the percentage of adequate literacy was relatively high among all medical departments (85–89%), revealing that almost one out of ten students had problematic levels in this domain. 

For Navigational Health Literacy, Public Health students had the highest percentage of adequate literacy (89.6%), and Nursing students had the lowest percentage (80.68%). The percentage of problematic cases was greater than that of other literacy types, with the highest percentage being 19.32% in Nursing. 

For Vaccination Health Literacy, Public Health students (93.6%) had the highest percentage of adequate literacy, followed by Nursing (90.91%), General Medicine (86.49%), Pharmacy (85.82%) and other fields (75.68%). The percentage of problematic patients was greater in the Other (24.32%) and Nursing students. (14.18%). 

Public Health students (Figure 1a) had the highest mean General Health Literacy (91.53 ± 13.22), followed by Nursing (91.05 ± 13.63), General Medicine (87.60 ± 19.02), and Other medical and health students (86.79 ± 18.75) and Pharmacy (85.19 ± 18.04). Overall, Public Health students consistently had the highest mean scores across all HL types, followed closely by Nursing students. Pharmacy representatives tended to have slightly lower mean scores than did those in the other departments. Similarly, the mean Digital and Navigational Health Literacy levels were similar across all medical departments. The Other health sciences students had a notably lower mean Vaccination Health Literacy score (79.73 ± 33.46) and a greater standard deviation, suggesting greater variability in scores. The ANOVA test results revealed a statistically significant variance in General Health Literacy levels across various educational programs. Subsequent Tukey’s post-hoc analysis demonstrated significant disparities in General Health Literacy scores between Pharmacy and Nursing (*p*-value = 0.03), as well as between Public Health and Pharmacy (*p*-value = 0.03). However, no statistically significant differences in General Health Literacy were observed among the remaining programs (Figure 1b). Based on the data presented in Figure 1b, significant group disparities are discerned when the 95% confidence interval excludes zero. This signifies that the p-value for these pairwise distinctions is <0.05.

### 3.4. Health Literacy Determinants

The determinants of General, Navigational, Digital, and Vaccination Health Literacy were analysed for all students. The results showed that several variables had a significant impact on these types of HL: 

For General Health Literacy, the variables that had a significant impact were financial ability to afford medical examination (hard: β = −4.41, *p* < 0.001), field of education (public health: β = 3.35, *p* < 0.05), age (β = 0.29, *p* < 0.001), living conditions (dormitory: β = 3.38, *p* < 0.01), Navigational Health literacy score (β = 0.13, *p* < 0.001), Vaccination Health literacy (β = 0.10, *p* < 0.001) and Digital Health literacy (β = 0.23, *p* < 0.001).

For Digital Health Literacy, area of origin (urban: β = 3.79, *p* < 0.001), financial ability to buy medication (easy: β = 2.88, *p* < 0.05), Navigational Health literacy score (β = 0.33, *p* < 0.001), Vaccination Health literacy (β = 0.13, *p* < 0.001) and General Health literacy (β = 0.34, *p* < 0.001) were found to be significant determinants. 

Navigational Health Literacy was significantly influenced by age (β = −0.34, *p* < 0.001), financial ability to afford medical examination (very hard: β = −10.86, p < 0.05), social status (β = 1.18, *p* < 0.01), health assessment (good: β = 12.85, *p* < 0.05), physical activity (3 times per week: β = 6.48, *p* < 0.01), Digital Health literacy score (β = 046, *p* < 0.001), Vaccination Health literacy (β = 0.23, *p* < 0.001) and General Health literacy (β = 0.26, *p* < 0.001). 

For the Vaccination Health Literacy of medical students, gender (male: β = −2.96, *p* = 0.05), living in a dormitory (β = −6.1, *p* < 0.01), age (β = 0.21, *p* < 0.05), field of education (Other: β = −10.01, *p* < 0.001), Digital Health literacy score (β = 021, *p* < 0.001), Navigational Health literacy (β = 0.25, *p* < 0.001) and General Health literacy (β = 0.23, *p* < 0.001) were found to be significant determinants.

## 4. Discussion

We conducted a comprehensive assessment of HL among medical and health students, including General, Digital, Navigational, and Vaccination Health Literacy. The findings provide valuable insights into the HL landscape among the study participants.

Our sample had a greater percentage of females, almost 70%. This is because there is a typical 1:3 ratio of males to females at medical higher education institutions in Kazakhstan [40]. It was shown in the research that female students tended to demonstrate higher levels of health literacy [41], which might in part explain the high overall levels of HL in this study.

Previous research has demonstrated the association of low HL with poor health status as well as the association of HL with health status [42,43]. The distribution of responses reflects the diverse natures of smoking, alcohol consumption, physical activity and healthy food habits within the population. Approximately 50% of the student population exhibits limited adherence to balanced nutritional intake, characterising a noteworthy prevalence of infrequent consumption of health-promoting food items. Another concerning finding is that a significant portion of our medical students appear to be inactive, warranting further investigation into potential implications for overall health and well-being.

Overall, the results of our study demonstrated that the students had high levels of General Health Literacy, as well as Digital, Navigational and Vaccination Literacy, with 95% of the sample exhibiting adequate General Health Literacy. The observed mean score for general HL was 88.26 (17.5), with a median of 91.7, which indicates a high level of General Health Literacy (Table 5). This finding is consistent with the results of other HL studies in students [44]. Furthermore, students in health-related study programs demonstrated higher levels of HL [45]. However, lower levels of HL among the student population were also identified in these studies [46]. While mean scores provide an overall view, the percentage breakdown helps identify specific areas of concern within the departments and fields of study. Targeted interventions might be necessary to address problematic HL.

In this study, Nursing and Public Health students demonstrated the highest general HL scores. According to the linear regression analysis, among the fields of study, only the Public Health and Nursing were positively and significantly associated with general HL scores. Notably, in Kazakhstan, Public Health study programs are carried out starting from the bachelor’s degree level (and further), which may explain in part why the students in this study program demonstrated the highest general HL scores. The results of other research showed that Nursing students demonstrated the lowest levels of HL [8]. According to the study by Birimoglu et al., the majority of nursing students (77.8%) had inadequate HL [47].

Notably, among the students in all educational programs in the medical universities, the Nursing students in this sample had the lowest percentage of adequate Navigational Health Literacy. Among all types of health literacy, Navigational Health Literacy was the most challenging, with 15.5% of the respondents in our sample having problematic levels. These findings are consistent with some of the previous related research and highlight the need for the development of tailored interventions in the future to address this issue [48].

It is worth noting that although the majority of the respondents in the sample demonstrated relatively high levels of HL, more than half of the respondents (53.4%) were involved in physical activity less than twice per week. Other studies conducted among adults indicate a positive relationship between HL and motivational preparedness for physical exercise, whereas physical inactivity and a sedentary lifestyle are associated with inadequate HL [30,49]. 

Furthermore, we investigated how various factors are associated with HL level. The results of the linear regression analysis showed that male gender was negatively associated with Vaccination Health Literacy only (β = −4.46, *p* < 0.05). Other research has also revealed that gender is related to the level of HL of students, but the conclusions of these studies vary. In particular, several studies have reported that female students have higher HL levels than male students [21,46,50,51]. In the study by Vozikis et al., HL was negatively associated with male gender [44]. In contrast, in the study by Sarhan et al., male respondents scored significantly higher than female respondents [52].

Age was positively and significantly correlated with the General Health Literacy score and Vaccination Literacy, which may be explained partly by the fact that older students have more encounters with the healthcare system, and more study and personal experience that may positively affect their HL. Upon evaluating potential predictors, Vashe’s study revealed that the age of individuals was a significant predictor of the total HL score. The younger participants had a reduced probability of achieving a desirable HL score overall compared to the older participants [53]. Similar results were found in our study; for example, the older the student was, the greater the level of general and Vaccination Health Literacy.

Financial disadvantage is the most powerful predictor of inadequate HL, followed by social status, education, age and gender [39]. In our study, difficulties in the financial ability to buy medicines and obtain medical information (very hard and hard) were negatively and significantly associated with all types of HL domains (General, Digital, Navigational and Vaccination), whereas the ease of accessing those expenses was positively associated with digital, navigational and vaccination literacies. Financial and social status (self-assessment) were positively and significantly associated with all types of HL, which is in accordance with the existing research demonstrating an association between income and HL [42,44].

Moreover, urban students had higher Digital Health Literacy levels than did rural students, in accordance with the findings of Zhang’s study, which was conducted in China [15]. Students residing in urban areas had greater HL than did those residing in smaller municipalities [54]. In addition, Griece’s study examined the Navigational Health Literacy of individuals with chronic illness in Germany and revealed a poorer level of Navigational Health Literacy. Our study revealed that having a history of chronic diseases was negatively associated with General Health Literacy levels [55].

There were notable disparities in HL based on exercise frequency in the study of Chu-Ko et al. Individuals with better HL scores tended to exercise more frequently on a weekly basis [56]. Kiran’s study revealed that students who engaged in physical exercise had a HL level that was 1.98 times greater than that of those who did not exercise [54]. A study conducted among university students in six major cities in Greece revealed that individuals who engaged in regular physical exercise exhibited HL levels that were 1.3 times greater than those who did not participate in exercise [44]. However, in our study, the occurrence of physical activity was significantly negatively associated with Digital HL and Navigational HL. According to the results of this study, alcohol consumption was negatively associated with Navigational HL. Unrecommendable alcohol consumption was linked to an increased likelihood of lacking adequate HL [42].

To our knowledge, this was the first comprehensive assessment of General, Digital, Navigational and Vaccination Literacy carried out among medical students in Kazakhstan via the short adapted forms of the HLS19-Q12 and optional instruments—Digital Health Literacy (HLS19-DIGI), Navigational Health Literacy (HLS19-NAV), and Vaccination Health Literacy (HLS19-VAC) [38]. Studies demonstrate that HL interventions and educational courses are effective at increasing different aspects of HL in medical students [57,58].

College students with lower HL reported more depressive symptoms. Universities can be vital in helping students receive early treatments that promote health, with a focus on HL and mental and physical health self-management. Early in their academic careers, medical college students could be given access to curricula that include mental health instruction [59]. HL programs should be tailored to better meet the requirements and characteristics of the various student populations in order to maximise their effectiveness [45].

### Limitations 

This study has several limitations. First, it is possible that only the most enthusiastic and motivated medical students completed the questionnaire, which may influence the skewed results and poor representativeness of the overall student population. 

A second limitation is that the majority of respondents were young students (median age is 19) and predominantly female. These results may not reflect the diversity of experience and knowledge of the student community as a whole, which may limit the generalisation of the results to older students and the male portion of the student population.

The descriptive aspect of the analytical methodology fails to account for multiple comparisons, potentially resulting in erroneous identification of significance where statistical significance is lacking.

## 5. Conclusions

The concept of HL goes far beyond the boundaries of healthcare settings and systems. The education sector can play a pivotal role in the process of disseminating health knowledge and enhancing the HL of future professionals, medical students and health students in non-health study programs. Medical and health students demonstrated relatively high overall levels of HL in this study. For General Health Literacy, the percentage of adequate literacy was the highest for Public Health, followed by Nursing students. The most challenging domain for all medical and health students was Navigational Health Literacy. Financial ability to buy medication and afford medical examination were significant determinants of all types of HL. Students’ HL should be evaluated on a regular basis in order to track their progress, identify potential gaps, and take relevant action. Since comprehensive HL instruction or interventions are still uncommon in the curricula, it seems reasonable to develop and incorporate appropriate courses for medical and health study programs.

## Figures and Tables

**Figure 1 healthcare-12-00907-f001:**
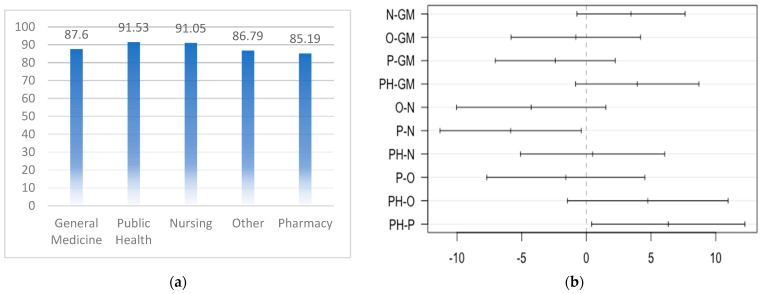
Medical and health students’ general health literacy levels: (**a**) Medical and health students’ General Health Literacy levels’ mean values; (**b**) Tukey’s post-hoc analysis results. GM—General Medicine, PH—Public Health, O—Other, P—Pharmacy, N—Nursing.

**Table 1 healthcare-12-00907-t001:** Socioeconomic sample characteristics.

Variable	n	General Medicine	Nursing	Pharmacy	Public Health	Other
Total	1042	496	176	134	125	111
Gender (male/female)	273/769	178/318	158/18	21/113	27/98	29/82
Age (mean/median)	22.2/19	19.2/19	30.4/27	21.0/20	22.6/20	22.8/19
Area of origin (urban/rural)	651/391	272/224	139/37	75/59	77/48	88/23
Living condition (rent/living with parents)	192/193	74/80	35/28	37/29	35/19	11/37
Living condition (dormitory/own apartment)	365/292	255/87	30/83	31/37	29/42	20/43
Father’s education (secondary/bachelor/professional/master degree/Ph.D.)	341/331/270/81/19	175/177/85/47/12	61/35/76/4/0	50/45/31/6/2	30/42/40/11/2	25/32/38/13/3
Mother’s education (secondary/bachelor/professional/master degree/Ph.D.)	271/395/225/136/15	127/218/63/76/12	56/39/72/9/0	39/55/24/13/3	25/46/33/21/0	24/37/33/17/0
Nuclear family (yes/no)	918/124	456/40	151/25	113/21	104/21	94/17
Financial ability to buy medication (easy/hard)	875/167	429/67	137/39	110/24	105/20	94/17
Financial ability to afford medical examination (easy/hard)	756/286	362/134	125/53	95/39	95/30	81/30
Social status (mean/median)	7.8/8	7.7/8	7.9/8	7.7/8	7.9/8	7.9/8
Financial status (mean/median)	7.1/7	7.3/7	6.7/7	6.7/7	7.4/7	7.2/7

**Table 2 healthcare-12-00907-t002:** Health-related sample information.

Variable	Total	Education Field				
		General Medicine	Nursing	Public Health	Pharmacy	Other
BMI (%)						
normal	632 (60.7)	324 (65.3)	94 (54.5)	74 (59.2)	76 (56.7)	64 (57.7)
underweight	134 (12.9)	74 (14.9)	14 (7.9)	9 (7.2)	27 (20.1)	10 (9.0)
overweight	133 (12.8)	46 (9.3)	42 (23.9)	23 (18.4)	9 (6.7)	13 (17.1)
obese	39 (3.7)	10 (2.0)	10 (5.7)	5 (4.0)	3 (2.2)	7 (6.3)
Health problems among family members (%)						
yes	306 (29.4)	151 (30.4)	47 (26.7)	49 (39.2)	34 (25.4)	25 (22.5)
no	738 (70.8)	345 (69.6)	129 (73.3)	76 (60.8)	100 (74.6)	86 (77.5)
Health assessment (%)						
good	776 (74.5)	392 (79.0)	115 (65.3)	90 (72)	96 (71.6)	83 (74.8)
bad	13 (1.2)	6 (1.2)	1 (0.6)	4 (3.2)	0 (0)	2 (1.8)
neither good nor bad	253 (24.3)	98 (19.8)	60 (34.1)	31 (24.8)	38 (28.4)	26 (23.4)
Activities limitations due to health problems (%)						
no activity limitations	333 (31.9)	141 (28.4)		38 (30.4)	56 (41.8)	37 (33.3)
no health problems	415 (39.8)	207 (41.7)		48 (38.3)	49 (36.6)	43 (38.7)
limited but not severely	294 (28.2)	148 (29.8)		39 (31.2)	29 (21.6)	31 (27.9)
Chronic diseases (%)						
yes	262	108 (21.8)	52 (29.5)	36 (28.8)	34 (25.4)	32 (28.8)
no	780	388 (78.2)	124 (70.5)	89 (71.2)	100 (74.6)	79 (71.2)

**Table 3 healthcare-12-00907-t003:** Health behaviour of the sample.

Period	Smoking (%)	Alcohol (%)	Physical Activity (%)	Healthy Nutrition (%)
Never	976 (93.7)	963 (92.4)	208 (19.9)	80 (7.7)
Less than one day per week	25 (2.4)	74 (7.1)	225 (21.6)	216 (20.7)
1 day	8 (0.8)	2 (0.2)	124 (11.9)	154 (14.8)
2 days	4 (0.4)	2 (0.2)	83 (7.9)	78 (7.5)
3 days	4 (0.4)	0	103 (9.9)	105 (10.1)
4 days	1 (0.09)	0	53 (5.1)	76 (7.3)
5 days	1 (0.09)	0	75 (7.2)	93 (8.9)
6 days	3 (0.3)	1 (0.09)	43 (4.1)	62 (5.9)
7 days	20 (1.9)	0	128 (12.3)	178 (17.1)

**Table 4 healthcare-12-00907-t004:** General, Digital, Navigational and Vaccination Health literacy assessment.

HL	Total	General Medicine	Public Health	Nursing	Pharmacy	Other
**Mean level, SD/Median**						
General	88.26, 17.5/91.7	87.60, 19.02/91.67	91.53, 13.22/100	91.05, 13.63/100	85.19, 18.04/91.67	86.79, 18.75/91.67
Digital	85.79, 22.8/100	86.27, 21.79/100	86, 23.24/100	85.79, 23.79/100	84.48, 25.79/100	84.48, 21.99/100
Navigational	84, 27.4/100	84.69, 26.81/100	86.39, 25.41/100	81.44, 29.04/100	85.01, 29.53/100	81.31, 27.14/100
Vaccination	87.9, 25.4/100	88.16, 24.35/100	91.80, 19.77/100	89.91, 24.23/100	87.50, 26.68/100	79.73, 33.46/100
**Adequate (%)/Problematic (%)**						
General	990 (95.01)/52 (5)	468 (94.45)/28 (5.65)	121 (96.8)/4 (3.2)	170 (96.59)/6 (3.40)	125 (93.28)/9 (6.72)	106 (95.49)/5 (4.50)
Digital	921 (88.39)/121 (11.61)	443 (89.31)/53 (10.69)	110 (88)/15 (12)	156 (88.64)/20 (11.36)	115 (85.82)/19 (14.18)	97 (87.39)/14 (12.61)
Navigational	881 (84.55)/161 (15.45)	423 (85.28)/73 (14.72)	112 (89.6)/13 (10.4)	142 (80.68)/34 (19.32)	113 (84.33)/21 (15.67)	91 (81.98)/20 (18.02)
Vaccination	905 (86.85)/137 (13.15)	429 (86.49)/67 (13.51)	117 (93.6)/8 (6.4)	160 (90.91)/16 (9.10)	115 (85.82)/19 (14.18)	84 (75.68)/27 (24.32)

**Table 5 healthcare-12-00907-t005:** Determinants of general, navigational, digital and vaccination health literacy.

Variable	General		Digital		Navigational		Vaccination	
	Estimates	*p*	Estimates	*p*	Estimates	*p*	Estimates	*p*
Intercept	34.79	<0.001	13.49	<0.001	14.86	0.06	36.18	<0.01
Gender (male)							−2.96	0.05
Age	0.29	<0.001			−0.34	<0.001	0.21	0.04
Area of origin (urban)			3.79	<0.001				
Living condition (dormitory)	3.38	<0.01					−6.11	<0.01
Financial ability to buy medication (easy)			2.88	0.04				
Financial ability to afford medical examination (very hard)					−10.86	0.02		
Financial ability to afford medical examination (hard)	−4.41	<0.001						
Social status					1.18	<0.01		
Field of education (Public health)	3.35	0.02						
Field of education (Other)							−10.01	<0.001
Health assessment (good)					12.85	0.03		
Smoking (5 days per week)							−92.43	<0.001
Physical activity (3 times per week)					6.48	0.03		
General Health Literacy			0.34	<0.001	0.26	<0.001	0.23	<0.001
Navigational Health Literacy	0.13	<0.001	0.33	<0.001			0.25	<0.001
Digital Health Literacy	0.23	<0.001			0.46	<0.001	0.21	<0.001
Vaccination Health Literacy	0.10	<0.001	0.13	<0.001	0.23	<0.001		
R^2^/Adjusted R^2^	0.39/0.37		0.45/0.45		0.48/0.47		0.32/0.31	

## Data Availability

The data presented in this study are available on request from the corresponding author.

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
