# Peer review of "General, Vaccination, Navigational and Digital Health Literacy of Students Enrolled in Different Medical and Health Educational Programs"

_healthcare, 2024, doi:10.3390/healthcare12090907_

Round 1

Reviewer 1 Report

Comments and Suggestions for Authors

Thank you for the opportunity to review this manuscript. As future health professionals and also as patients, students should be aware of the need for empowerment in health care. Please find my suggestions.

ABSTRACT

-Please describe de date of the study.

-Lines 21-23: “The Public health students exhibited the highest General Health Literacy (91.53±13.22), followed by the Nursing, General medicine, Pharmacy, and students of other educational programs

·                     Please follow the order of the figure 1.

·                    In the abstract the authors could write who are the “other students”

INTRODUCTION

-Lines 40-41: “The WHO has identified the problem of rising HL in the population as an important public health concern.”  Has the WHO identified the problem of increased LH or a lack of HL?

-Lines 74-75: “The instrument showed an excellent level of consistency across all participating countries, and the unidimensionality of the scale was confirmed by CFA and Rasch models…”  Please describe CFA.

MATERIAL AND METHODS

-Lines 85-86: “A cross-sectional survey was conducted between September and November of 2023 in a representative sample of 1042 students of various educational programs from three medical universities…”  Is it correct to describe medical universities when other health professionals are involved?

-Line 90: “The survey was available on print and online.” Where was the survey available online?

-Line 91: “The study involved students from the first to the fifth years of study.” It could be: The study involved students from the first to the fifth years of the universities.

-Is it HLS19-Q12 (Line 71) or HLS19-12 (Line 15/ Line 97)? Please correct all the text.

-Lines 101-103: “Additionally, optional instruments, such as Digital Health Literacy (HLS19-DIGI), Navigational Health Literacy (HLS19-NAV), and Vaccination Health Literacy (HLS19-VAC), were used…” Please, when describing the questionnaires, follow the order of the questionnaires according to this paragraph.

-What about HLS19-Q12? How many questions are there? Please describe.

-Lines 103-104: “A total of 31 correlates were employed for all the surveys, with the HLS19-DIGI and HLS19-VAC utilizing additional correlates.” Please describe this information further.

-Line 124: “The Local Bioethics Commission of the Medical University of Karaganda, dated October 11, 2022 (Protocol 1), approved the study.” What about the other two universities?

-Lines 145-147: “The criteria for classifying HL were determined as follows: less than 50 years (inadequate); between 50 and 66.66 years (problematic); between 66.67 and 83.33 years (adequate); and above 83.34 years (excellent).”  Years? Is it correct?

-Line 166: “…1/3 of fathers and mothers.” Please write in % like the other descriptions.

-Line 180: Please describe BMI.

-Line 183: “…the majority of the patients were in the normal weight category…”  Patients or students? Please correct this information.

Lines 189-190. “Moreover, only one-third of all the students reported no activity limitations due to health problems”.  Please write in % like the other descriptions.

Table 2. This table could be better presented. It is hard to read.

Table 3. Was it considered the number of smoking days and not the smoking burden (Number of cigarettes x frequency of smoking)?

Lines 221-222. “In the assessment of HL across different fields, the following conclusions were drawn…”  In this Result section, please do not write “conclusions”.

Lines 238-240. “Public health students (Figure 1) had the highest mean General Health Literacy (91.53±13.22), followed by Nursing (91.05±13.63), General medicine (87.60±19.02), Pharmacy (85.19±18.04), and Other medical students (86.79±18.75).” Please follow the order of the % like the figure 1.

DISCUSSION

-Lines 278-279: “We conducted a comprehensive assessment of HL among medical students, including General, Digital, Navigational, and Vaccination Health Literacy.” Is it correct to describe medical students when other health students are involved?

-Lines 281-283. “Our sample had a greater percentage of females, almost 70%. This is because there is a typical 1:3 ratio of males to females at medical higher education institutions in Kazakhstan” Why? Are there more females in population? The authors could explore whether or not this gender gap could be an advantage.

-Line 290: “…our medical students appear to be inactive…” Is it correct to describe medical students when other health students are involved?

-Lines 314-315: “Notably, among the students in all educational programs in the medical universities, the Nursing students in this sample had the lowest percentage of adequate Navigation Health Literacy.” Is it correct to describe medical universities when other health students are involved?

CONCLUSION

-Line 400: Is it correct to describe medical universities when other health students are involved?

Author Response

Response to Reviewers

The authors thank the reviewers for the time and effort spent to improve the manuscript. The efforts are greatly acknowledged. We agree in general with all points raised by the reviewers and have revised the manuscript accordingly. We believe that the improved manuscript will be a valuable addition of knowledge to the journal regarding health literacy level among students. Below follow all comments from the reviewers and a description of actions taken. The reviewer´s text is in black and authors´ replies are in blue.

Response to Reviewer 1 Comments

  1. ABSTRACT

Please describe de date of the study.

Thank you for your comment. We agreed with it and added the date of study in the part “Abstract”.

  1. Lines 21-23: “The Public health students exhibited the highest General Health Literacy (91.53±13.22), followed by the Nursing, General medicine, Pharmacy, and students of other educational programs”.

Please follow the order of the figure 1.

Thank you for your note. We changed this line, and it is stated as followed:

“The Public health students exhibited the highest General Health Literacy (91.53±13.22), followed by the Nursing, General medicine, students of other educational programs (Dentistry and Biolmedicine) and Pharmacy.”

  1. Lines 23-24: “The Public health students exhibited the highest General Health Literacy (91.53±13.22), followed by the Nursing, General medicine, students of other educational programs (Dentistry and Biomedicine) and Pharmacy”

In the abstract the authors could write who are the “other students”

Thank you for your note. We added in the abstract “other students”, and it is stated as followed:

“The Public health students exhibited the highest General Health Literacy (91.53±13.22), followed by the Nursing, General medicine, students of other educational programs (Dentistry and Biomedicine) and Pharmacy.”

  1. INTRODUCTION

-Lines 40-41: “The WHO has identified the problem of rising HL in the population as an important public health concern.”  Has the WHO identified the problem of increased LH or a lack of HL?

Thank you for your note. The WHO has identified the problem of limited HL in the population as an important public health concern

  1. Lines 74-75: “The instrument showed an excellent level of consistency across all participating countries, and the unidimensionality of the scale was confirmed by CFA and Rasch models…”  Please describe CFA.

Thank you for the comment. In the manuscript, we described CFA as a confirmatory factor analysis (lines 83-84).

  1. Lines 85-86: “A cross-sectional survey was conducted between September and November of 2023 in a representative sample of 1042 students of various educational programs from three medicaluniversities…”  Is it correct to describe medical universities when other health professionals are involved?

Thank you for your question. Medical universities in Kazakhstan prepare graduates in medical and health-related educational programs such as Public health, Biomedicine, Pharmacy, Nursing, and Dentistry.  In this research, we addressed health literacy among students enrolled in various medical and health-related educational programs taught at three medical universities in Kazakhstan. Students of Biomedicine and Dentistry were combined into the group “Other students” for the sake of convenience during the statistical analysis. However, “Other” students are still the students of those medical universities enrolled in the Biomedicine and Dentistry educational programs and not the students of the non-health educational programs.

  1. Line 90: “The survey was available on print and online.” Where was the survey available online?

Thank you. In the manuscript, we added the next sentence: “The electronic version of the questionnaire was available online on the Google platform, with access offered via QR code.” (lines 100-101)

  1. Line 91: “The study involved students from the first to the fifth years of study.” It could be: The study involved students from the first to the fifth years of the universities.

Thank you. We corrected in the manuscript (Line 102).

  1. Is it HLS19-Q12 (Line 71) or HLS19-12 (Line 15/ Line 97)? Please correct all the text.

Thank you. We corrected all the text of the manuscript.

  1. Lines 101-103: “Additionally, optional instruments, such as Digital Health Literacy (HLS19-DIGI), Navigational Health Literacy (HLS19-NAV), and Vaccination Health Literacy (HLS19-VAC), were used…” Please, when describing the questionnaires, follow the order of the questionnaires according to this paragraph.

Thank you. In the “Materials and Methods” part we changed the order of the questionnaire descriptions (lines 116-132).

  1. What about HLS19-Q12? How many questions are there? Please describe.

Thank you. We added the next sentence, and, in the manuscript, it is followed as “It is 12-item short-form questionnaire that belongs to the HLS19 family of HL measurement tools” (lines 109-110).

  1. Lines 103-104: “A total of 31 correlateswere employed for all the surveys, with the HLS19-DIGI and HLS19-VAC utilizing additional correlates.” Please describe this information further.

Thank you. It was corrected. Correlates are described in the first passage of the “Measures and Statistical Analysis” part (lines 149-156).

  1. Line 124: “The Local Bioethics Commission of the Medical University of Karaganda, dated October 11, 2022 (Protocol 1), approved the study.” What about the other two universities?

Thank you for your question. As the research grant is held by the Medical University of Karaganda, where the Project was carried out, obtaining a decision from the Bioethics Commission of the same institution was a need. The MUK Bioethics Commission's decision is  valid for all participants in the study

  1. Lines 145-147: “The criteria for classifying HL were determined as follows: less than..50 years(inadequate); between 50 and 66.66 years(problematic); between 66.67 and 83.33 years (adequate); and above 83.34 years (excellent).”  Years? Is it correct?

Thank you. We changed years to the points in the manuscript (lines 164-166).

  1. Line 166: “…1/3of fathers and mothers.” Please write in % like the other descriptions.

Thank you. It was rewritten in the manuscript as follows: “In terms of parents’ education level, 31.7% of fathers and 37.9% of mothers had at least a bachelor's degree.”

  1. Line 180: Please describe BMI.

Thank you for the comment. In the manuscript, we described BMI as a body mass index (lines 198-199).

  1. Line 183: “…the majority of the patientswere in the normal weight category…”  Patients or students? Please correct this information.

Thank you for the comment. In the manuscript, we changed patients to students (line 201).

  1. Lines 189-190. “Moreover, only one-thirdof all the students reported no activity limitations due to health problems”.  Please write in % like the other descriptions.

Thank you. It was rewritten in the manuscript as follows: “Moreover, only 31.9% of all the students reported no activity limitations due to health problems.”

  1. Table 2. This table could be better presented. It is hard to read.

Thanks for your suggestion. We modified this table in the text of the manuscript, and it is stated as follows:

Variable  

Total  

Education field  

General medicine 

Nursing 

Public health 

Pharmacy 

Other 

BMI (%) 

normal

632 (60.7)

324 (65.3)

94 (54.5)

74 (59.2)

76 (56.7)

64 (57.7)

under

weight

134 (12.9)

74 (14.9)

14 (7.9)

9 (7.2)

27 (20.1)

10 (9.0)

overweight

133 (12.8)

46 (9.3)

42 (23.9)

23 (18.4)

9 (6.7)

13 (17.1)

obese

39 (3.7)

10 (2.0)

10 (5.7)

5 (4.0)

3 (2.2)

7 (6.3)

Health problems among family members (%)

yes

306 (29.4)

151 (30.4)

47 (26.7)

49 (39.2)

34 (25.4)

25 (22.5)

no 

738 (70.8)

345 (69.6)

129 (73.3)

76 (60.8)

100 (74.6)

86 (77.5)

Health assessment (%)  

good

776 (74.5)

392 (79.0)

115 (65.3)

90 (72)

96 (71.6)

83 (74.8)

bad

13 (1.2)

6 (1.2)

1 (0.6)

4 (3.2)

0 (0)

2 (1.8)

neither good nor bad

253 (24.3)

98 (19.8)

60 (34.1)

31 (24.8)

38 (28.4)

26 (23.4)

Activities limitations due to health problems (%)

no activity limitations

333 (31.9)

141 (28.4)

38 (30.4)

56 (41.8)

37 (33.3)

no health problems

415 (39.8)

207 (41.7)

48 (38.3)

49 (36.6)

43 (38.7)

limited but not severely

294 (28.2)

148 (29.8)

39 (31.2)

29 (21.6)

31 (27.9)

Chronic diseases (%)

yes

262

108 (21.8)

52 (29.5)

36 (28.8)

34 (25.4)

32 (28.8)

 no  

780

388 (78.2)

124 (70.5)

89 (71.2)

100 (74.6)

79 (71.2)

  1. Table 3. Was it considered the number of smoking days and not the smoking burden (Number of cigarettes x frequency of smoking)?

We agree with your observation. However, it's important to clarify that the questionnaire utilized in our study was obtained from the M-Pohl, and thus, we were unable to modify the specific questions within the questionnaire.

  1. Lines 221-222. “In the assessment of HL across different fields, the following conclusionswere drawn…”  In this Result section, please do not write “conclusions”.

Thank you very much. We removed this sentence from the manuscript.

  1. Lines 238-240. “Public health students (Figure 1) had the highest mean General Health Literacy (91.53±13.22), followed by Nursing (91.05±13.63), General medicine (87.60±19.02), Pharmacy (85.19±18.04), and Other medical students (86.79±18.75).” Please follow the order of the % like the figure 1.

Thank you. We have corrected it in the correct order in the text of the Manuscript (lines 258-260).

23.DISCUSSION

-Lines 278-279: “We conducted a comprehensive assessment of HL among medical students, including General, Digital, Navigational, and Vaccination Health Literacy.” Is it correct to describe medical students when other health students are involved?

Thank you for your question. We understand your question and let us to provide an explanation of the education system in Kazakhstan. Graduates of medical and health-related educational programs, including public health, Biomedicine, pharmacy, nursing, and dentistry, are prepared by Kazakhstan's medical universities.  In this study, health literacy among students enrolled in different medical and health-related educational programs taught at three Kazakhstani medical universities was investigated. For the purpose of simplicity during the statistical analysis, students studying Biomedicine and dentistry were grouped into one group, "Other students." In contrast, "Other" students are not those engaged in non-healthy educational programs; rather, they are still students at those medical universities studying Biomedicine and dentistry. All the students of medical university referred to as “medical students” in this study.

  1. Lines 281-283. “Our sample had a greater percentage of females, almost 70%. This is because there is a typical 1:3 ratio of males to females at medical higher education institutions in Kazakhstan” Why? Are there more females in population? The authors could explore whether or not this gender gap could be an advantage.

Thank you for your comment. You are correct to highlight the gender disparity in our sample. The predominance of females is characteristic of medical higher education institutions in Kazakhstan, but not reflective of the general population where distribution is more balanced. We added the reference showing that indeed the ratio is 1/3. Trends in Higher Education in Kazakhstan: Which Majors Are Popular Among Men and Women? https://qazmonitor.com/news/2295/trends-in-higher-education-in-kazakhstan-which-majors-are-popular-among-men-and-women

  1. Line 290: “…our medical students appear to be inactive…” Is it correct to describe medical students when other health students are involved?

 Thank you for your question. We understand your question and let us to provide an explanation of the education system in Kazakhstan. Graduates of medical and health-related educational programs, including public health, Biomedicine, pharmacy, nursing, and dentistry, are prepared by Kazakhstan's medical universities.  In this study, health literacy among students enrolled in different medical and health-related educational programs taught at three Kazakhstani medical universities was investigated. For the purpose of simplicity during the statistical analysis, students studying Biomedicine and dentistry were grouped into one group, "Other students." In contrast, "Other" students are not those engaged in non-healthy educational programs; rather, they are still students at those medical universities studying Biomedicine and dentistry. All students at medical university referred to as “medical students” in this study.

  1. Lines 314-315: “Notably, among the students in all educational programs in the medical universities, the Nursing students in this sample had the lowest percentage of adequate Navigation Health Literacy.” Is it correct to describe medical universities when other health students are involved?

Thank you for your question. We understand your question and let us to provide an explanation of the education system in Kazakhstan. Graduates of medical and health-related educational programs, including public health, Biomedicine, pharmacy, nursing, and dentistry, are prepared by Kazakhstan's medical universities.  In this study, health literacy among students enrolled in different medical and health-related educational programs taught at three Kazakhstani medical universities was investigated. For the purpose of simplicity during the statistical analysis, students studying Biomedicine and dentistry were grouped into one group, "Other students." In contrast, "Other" students are not those engaged in non-healthy educational programs; rather, they are still students at those medical universities studying Biomedicine and dentistry. All students at medical university referred to as “medical students” in this study.

  1. CONCLUSION

-Line 400: Is it correct to describe medical universities when other health students are involved?

Thank you for your question. We understand your question and let us to provide an explanation of the education system in Kazakhstan. Graduates of medical and health-related educational programs, including public health, Biomedicine, pharmacy, nursing, and dentistry, are prepared by Kazakhstan's medical universities.  In this study, health literacy among students enrolled in different medical and health-related educational programs taught at three Kazakhstani medical universities was investigated. For the purpose of simplicity during the statistical analysis, students studying Biomedicine and dentistry were grouped into one group, "Other students." In contrast, "Other" students are not those engaged in non-healthy educational programs; rather, they are still students at those medical universities studying Biomedicine and dentistry. All students at medical university referred to as “medical students” in this study.

Submission Date

25 March 2024

Date of this review

11 Apr 2024

Reviewer 2 Report

Comments and Suggestions for Authors

Based on which criteria was the sample size decided? How were the students participating in the research selected in educational institutions? (Simple random, systematic? random?, convenience sampling?

The dependent variable of the research is Health Literacy. Is it taken as a dependent variable in other scales? ( Digital, Navigational, Vaccination ) . How might other scales affect the results if they were taken as independent variables?

In Table 2, the total BMI classification value does not exceed the number 1042.

Table 5 makes it difficult to understand. It can be seen that almost all independent variables are included in all 4 scales.

More clear results should be given by choosing stepwise regression analysis.

When the means and standard deviations of the scales are examined in Table 4; A coefficient of variation (CV) ranging between 25 and 30 is calculated. Since the Total Score is constantly variable, has it been tested for conformity with the Normal Distribution?

Since the scale questions are Likert-type scales, shouldn't the total score be considered as an ordinal measurement level?

Author Response

Response to Reviewer 2. Comments and Suggestions for Authors

Comment 1 Based on which criteria was the sample size decided? How were the students participating in the research selected in educational institutions? (Simple random, systematic? random? convenience sampling?

Response 1: Thank you for your insightful comment. In our study, the sample size of 1042 students was determined based on the total population of students at the three medical universities in Karaganda, Astana and Almaty cities. The goal was to ensure that the sample size is large enough to provide statistically significant results and reflective of the broader student body across these institutions.

We used the stratified sampling method to select participants, ensuring representation from various educational programs such as General Medicine, Public Health, Nursing, Pharmacy, Dentistry, and Biomedicine. Stratified sampling involves dividing the total population into distinct subgroups or 'strata' (in our case, based on educational programs) and then randomly selecting participants from each subgroup. This method helps to reduce sampling error and ensures that specific groups are not underrepresented.

Within each stratum (educational program), participants were selected using simple random sampling. This was done through randomized number generation corresponding to student enrollment lists.

Comment 2 The dependent variable of the research is Health Literacy. Is it taken as a dependent variable in other scales? (Digital, Navigational, Vaccination) . How might other scales affect the results if they were taken as independent variables?

Response 2: We formed 4 Stepwise Linear regression models, in each of which the dependent variables were General, Navigation, Digital and Vaccination Health Literacy. Also, General, Navigation, Digital and Vaccination Health Literacy was considered as an independent variable in each of the scales. The independent variables are presented and described in Table 1, Table 2. Table 3, and Table 4.

Comment 3 In Table 2, the total BMI classification value does not exceed the number 1042.

The total BMI classification value does not reach 1042, because not all respondents indicated height and weight parameters during the survey.

Comment 4 Table 5 makes it difficult to understand. It can be seen that almost all independent variables are included in all 4 scales.

Respond 4: Thank you for your comment. We have optimized the table and now it looks like this:

Variable  

General  

Digital  

Navigational  

Vaccination  

Estimates 

p   

Estimates 

p  

Estimates 

p  

Estimates 

p  

Intercept

34.79

<0.001

13.49

<0.001

14.86

0.06

36.18

<0.01

Gender (male)  

-2.96 

0.05  

Age  

0.29  

<0.001 

-0.34  

<0.001  

0.21  

0.04  

Area of origin (urban)  

3.79  

<0.001  

Living condition (dormitory)  

3.38  

<0.01 

-6.11  

<0.01  

Financial ability to buy medication(easy) 

2.88

0.04

Financial ability to afford medical examination (very hard)  

-10.86  

0.02  

Financial ability to afford medical examination (hard)  

-4.41 

<0.001 

Social status  

1.18  

<0.01  

Field of education (Public health)  

  3.35

0.02  

Field of education (Other)  

-10.01  

<0.001  

Health assessment (good)

12.85

0.03

Smoking (5 days per week)  

-92.43  

<0.001  

Physical activity (3 times per week)  

6.48  

0.03  

General Health Literacy

0.34

<0.001

0.26

<0.001

0.23

<0.001

Navigational Health Literacy

0.13

<0.001

0.33

<0.001

0..25

<0.001

Digital Health Literacy

0.23

<0.001

0.46

<0.001

0.21

<0.001

Vaccination Health Literacy

0.10

<0.001

0.13

<0.001

0.23

<0.001

R2/Adjusted R2

0.39/0.37

0.45/0.45

0.48/0.47

0.32/0.31

Comment 5 More clear results should be given by choosing stepwise regression analysis.

Response 5: Thank you for your recomendation. Both-direction Stepwise Linear regression was perfomed to determine factors influencing General, Digital, Vaccination and Navigational Health literacy. The dependent variables in the models were General, Vaccination, Digital and Navigational Health literacy. The independent variables are presented and described in Table 1, Table 2. Table 3, and Table 4. By adding important variables and eliminating unimportant ones, the method in both-direction stepwise regression optimizes the model by combining forward and backward stages. The optimized regression models in this study are as follows:

General Health Literacy = Age + Living conditions + Financial ability to afford medical examination + Health assessment + Digital Health Literacy Score + Navigational Health Literacy Score + Vaccination Health Literacy Score + Field of Medical Education.

Digital Health Literacy = Area of origin + Financial ability to afford medical examination + General Health Literacy Score + Navigational Health Literacy Score + Vaccination Health Literacy Score.

Navigational Health Literacy = Age + Financial ability to afford medical examination + Social status + Health assessment + General Health Literacy Score + Digital Health Literacy Score + Vaccination Health Literacy Score.

Vaccination Health Literacy = Gender + Age + Living condition + Smoking + General Health Literacy Score + Digital Health Literacy Score + Navigational Health Literacy + Field of Medical Education.

Comment 6 When the means and standard deviations of the scales are examined in Table 4; A coefficient of variation (CV) ranging between 25 and 30 is calculated. Since the Total Score is constantly variable, has it been tested for conformity with the Normal Distribution?

Respond 6: Thank you for your question. In Table 4 we tested Total Score for conformity with the Normal Distribution using Shapiro-Wilk test (p-value<0.001).

Comment 7 Since the scale questions are Likert-type scales, shouldn't the total score be considered as an ordinal measurement level?

Response 7: Yes, it is true that individual Likert scale items are ordinal in nature, the sum or mean of a collection of Likert-scale items is often treated as continuous under the assumption of an interval scale. This is particularly common when using multiple Likert items to construct a composite measure like a scale or index, as in our study.

Round 2

Reviewer 2 Report

Comments and Suggestions for Authors

The corrections made to the article by the author are sufficient. Acceptable.

Author Response

Thanks